# Synergetic Effects of Silver Nanowires and Graphene Oxide on Thermal Conductivity of Epoxy Composites

**DOI:** 10.3390/nano9091264

**Published:** 2019-09-05

**Authors:** Li Zhang, Wenfeng Zhu, Ying Huang, Shuhua Qi

**Affiliations:** Department of Applied Chemistry, School of Natural and Applied Sciences, Northwestern Polytechnical University, Xi’an 710072, China (L.Z.) (W.Z.) (Y.H.)

**Keywords:** silver nanowires, graphene oxide, synergetic effects, epoxy composite

## Abstract

One-dimensional silver nanowires (AgNWs) and two-dimensional graphene oxide (GO) were combined to construct a three-dimensional network structure. The AgNWs can effectively inhibit stacking of adjacent GO sheets by occupying regions between layers of GO. Moreover, the GO sheets embedded in the gaps of the AgNWs network increase the interfacial contact area between the AgNWs and the epoxy matrix, resulting in the formation of more efficient phonon transport channels. To prepare an epoxy-based thermal conductive composite, hybrid networks were fabricated and added to epoxy resin using a solution mixing method. Significant synergistic effects were observed between the AgNWs and GO sheets. The thermal conductivity of epoxy composites filled with 10 wt.% AgNW/GO hybrids was found to be 1.2 W/mK and the impact strength was 28.85 KJ/m^2^, which are higher than the corresponding values of composites containing AgNWs or GO sheets alone. Thus, the thermal conductivity and impact strength of the epoxy composites were improved. The additive effects are mainly owing to the improved interfacial contact between the hybrid fillers and the epoxy resin, resulting in a more efficient phonon transport network. The use of hybrid fillers with different structures is a simple and scalable strategy for manufacturing high-performance thermally conductive materials for electronic packaging.

## 1. Introduction

With the high integration of electronic components in modern chip packaging industry, electronic devices now generate an enormous amount of heat during operation, making it necessary to improve thermal interface materials to solve this problem [1,2,3,4]. Operating temperature affects the stability of electronic equipment. Moreover, cumulative heat will directly affect stability and lifespan, and could lead to other serious consequences [5,6]. Therefore, rapid dissipation of heat generated by electronic devices is crucial to ensure normal operation. In typical integrated circuit package, heat spreaders and heat sinks are used to dissipate heat generated by electronic components. However, the gap between the contact surfaces of components reduces the effective thermal conductivity. Usually, these gaps are filled with thermal interface materials (TIMs) to reduce the thermal contact resistance [7,8,9,10,11]. As such, many researchers have focused on developing polymer-based thermal interface materials. Polymeric composites used as thermal interface materials are lightweight and have low thermal expansion coefficients, low dielectric constants, and low dielectric losses, and therefore, offer a number of advantages. Polymers filled with thermally conductive particles such as aluminum oxide (Al_2_O_3_) [12,13,14], boron nitride (BN) [15,16,17], aluminum nitride (AlN) [18,19,20], and silicon carbide (SiC) [21,22,23] have been used to enhance the performance of polymer composites. In general, extremely high loading of fillers (about 30–60 vol.%) can increase the thermal conductivity of polymer composites, but can affect mechanical properties and presents processing difficulties.

In the field of electronic packaging, although many polymers are used as matrix materials for the preparation of thermally conductive composites, the most widely used are epoxy resins (EP). Epoxy resin is a thermosetting resin containing two or more reactive epoxy groups in the molecule. Epoxy resin has excellent adhesion, wear resistance, mechanical properties, electrical insulation properties, and chemical stability. Epoxy resin is easy to process and cost-effective, and has been widely used in electronic packaging [24,25]. Based on the above advantages, epoxy resin was used as a matrix material in this article.

Recently, one-dimensional (1-D) silver nanowires (AgNWs) have emerged as promising fillers for thermally conductive materials due to their high thermal conductivity and large specific surface area. According to related reports, the thermal conductivity of AgNWs is in the range of 200~250 W/mK. When the system size of AgNWs is close to or smaller than the electron mean free path, there is electron scattering at the grain boundaries and surfaces. Therefore, for AgNW, its thermal conduction mainly depends on electrons, but phonons also contribute to thermal conduction [26,27]. Composites based on AgNWs exhibit superior thermal conductivity performance. For example, Chen et al. [28] prepared high thermal conductivity composites by blending modified AgNWs with epoxy resin. The thermal conductivity coefficient of the composite, containing 4 vol.% of modified AgNWs, was shown to reach 1.03 W/mK. However, AgNWs are expensive to manufacture and their composites have poor mechanical strength, which limits their large-scale application. Therefore, it is necessary to introduce another inexpensive filler to improve the thermal conductivity of the composite while enhancing the mechanical properties.

Carbon-based nanofillers have been widely used in the preparation of TIMs due to their high thermal conductivity. For example, carbon nanotubes (2000~6000 W/mK) and graphene (2000~5300 W/mK) were added to the polymer matrix to improve the thermal conductivity of the composite, which has been reported in many literatures [29,30,31,32]. However, carbon nanotubes and graphene also have high electrical conductivity and are expensive, which is disadvantageous for TIMs. Graphene oxide (GO) has a rather low thermal conductivity (0.14~2.87 W/mK) [33,34] compared to these carbon nanofillers, but it is still used in the preparation of thermally conductive materials because of its low cost and mass production capabilities. In addition, two-dimensional (2-D) GO can easily form a strong interaction with the resin matrix due to the presence of a large number of hydroxyl and carboxyl polar groups on its surface [35,36]. Furthermore, the presence of polar groups on GO can improve the mechanical properties of nanocomposites and decrease the percolation threshold of fillers in polymer matrix. Table 1 shows the thermal conductivities of epoxy and fillers. The potential use of GO for thermal management and enhancing thermal conductivity of polymer-based nanocomposites has been demonstrated. Tseng and coworkers [2] dispersed chemically functionalized GO into a polyimide matrix, which was shown to increase the thermal conductivity of resin containing 10 wt.% GO to 0.81 W/mK. 

The use of hybrid fillers can effectively increase the thermal conductivity of composites, which has been reported in many literatures. For example, Barani et al. [37] prepared an epoxy composite filled with graphene and copper nanoparticles with a thermal conductivity of 13.5 W/mK. By adjusting the relative content of graphene, the thermal conductivity of the composite exhibits a completely different relationship with the filler content. Lewis et al. [38] investigated the thermal and electrical properties of hybrid epoxy composites with graphene and boron nitride fillers. The epoxy composites they prepared had a thermal conductivity of 6.5 W/mK, and the electrical conductivity of the composites was found to vary drastically as the relative content of boron nitride increased. However, there are few reports on AgNW/GO hybrids in the study of thermal conductivity materials. Therefore, the hybrid filler composed of 1-D AgNWs and 2-D GO nanosheets was used to effectively improve the thermal conductivity and mechanical properties of epoxy resin composites. In such a hybrid filler system, elongated AgNWs can enter adjacent graphene oxide sheets to inhibit their agglomeration, thereby effectively increasing the contact area between the hybrid filler and the epoxy resin. Epoxy composites containing AgNW/GO hybrid fillers prepared by a solution mixing method demonstrate improved thermal conductivity and mechanical strength compared to epoxy composites prepared with AgNWs or GO alone. The morphology and structure of AgNW/GO hybrid fillers and their epoxy composites were investigated and the synergetic effects of AgNW/GO hybrid fillers and their reinforcing mechanism are discussed.

## 2. Experimental

### 2.1. Materials

Natural graphite powder (size <25 μm, purity >99.9%) was purchased from Qingdao Xinghe Graphite Co., Ltd., Qingdao, China. Sodium nitrate (NaNO_3_), sulfuric acid (H_2_SO_4_; 98%), hydrochloric acid (HCl; 37%), hydrogen peroxide (H_2_O_2_; 35%), and potassium permanganate (KMnO_4_) were purchased from Guangdong Guanghua chemical Co., Ltd., Shantou, China. Ethanol, glycerol, and sodium chloride (NaCl) were purchased from Tianjin Fuyu Chemical Co., Ltd., Tianjin, China. Poly-vinylpyrrolidone (PVP) with an average molecular weight of 400,000 was purchased from Tokyo Chemical Industry Co., Ltd., Tokyo, Japan. Silver nitrate (AgNO_3_; 99.5%) was purchased from Aladdin Bio-Chem Technology Co., Ltd., Shanghai, China. Epoxy (E-51; MW: 370~420, epoxide equivalent: 190) was supplied by Zhejiang Materials Industry Chemical Group Co., Ltd., Jinhua, China. The molecular structure of epoxy E51 was shown in Figure 1. Finally, m-xylylenediamine (MXDA) was obtained from Shanghai Macklin Chemistry Co., Ltd., Shanghai, China and used as the curing agent.

### 2.2. Preparation of Silver Nanowires

AgNWs were synthesized by reducing AgNO_3_ with the polyol in a solution containing a polymeric surfactant [39]. Briefly, 5.85 g PVP and 190 mL glycerol were added to a 500 mL round bottom flask, the temperature was controlled at 50 °C and mechanical agitation was used to ensure complete dispersion of PVP. Further heating was stopped after the PVP powder was completely mixed with glycerol. When the solution was completely cooled, 1.58 g of AgNO_3_ was added thereto. Then, 59 mg of NaCl was dissolved in a mixed solution containing 0.5 mL H_2_O and 10 mL glycerol, and added to the flask together. The solution was heated to 210 °C and reacted at this temperature for one hour with gentle agitation during the process. When the reaction was completed, further heating was stopped and the solution cooled down to room temperature. Finally, 200 mL H_2_O was added to the solution and the mixture was centrifuged at 8000 rpm for 30 min. The obtained AgNW was washed with ethanol and centrifuged three times to remove residual PVP on the surface, and then the collected AgNW was dispersed in an ethanol solution for later use. The SEM photograph of synthesized AgNWs is shown in Figure 2a.

### 2.3. Preparation of Graphene Oxide Nanosheets

GO sheets were prepared from graphite powder by the modified Hummers’ method [40]. Briefly, 1 g graphite powder and 1 g NaNO_3_ were added to 46 mL concentrated H_2_SO_4_ in a 250 mL three-neck flask. The three-necked flask was placed in an ice bath and stirred for 4 h to homogeneously disperse the graphite in the mixture. Then, 6 g KMnO_4_ was slowly added to the flask within 1 h in an ice bath. The flask containing the mixture was then placed in an oil bath and stirred at 25 °C for 6 h. After the reaction, 100 mL deionized (DI) water was added to the mixture, and the solution was distilled and heated to 60 °C. Then, 200 mL DI water containing 20% H_2_O_2_ was added to the mixture to reduce residual oxidant. The mixture was then filtered and washed several times with 5% HCl solution and DI water. The graphite oxide was dispersed in DI water to obtain a brown suspension, which was dispersed under ultrasonication for 6 h. The mixture was then centrifuged (6000 rpm, 40 min) to obtain exfoliated GO nanosheets. The GO nanosheets were then dispersed in ethanol under ultrasonication for 2 h for later use. The representative morphology of synthesized GO sheets is shown in Figure 2b.

### 2.4. Fabrication of Composite Material

In this paper, solvent mixing method was used to prepare composite materials. First, AgNWs and GO nanosheets dispersed in ethanol were mixed together and sonicated for 20 min to facilitate the AgNW/GO hybrid assembly process. Relative proportions of the two fillers were adjusted by varying the AgNW and GO content. The mixture was heated to 70 °C, and then epoxy resin was slowly added to the solution and sonicated for 20 min to allow sufficient mixing of the hybrid filler and epoxy. The resulting mixture was placed in a vacuum oven at 70 °C to remove the ethanol. The curing agent (MXDA) was then added to the epoxy composites and allowed to mix for 10 min under ultrasonication. The relative ratio of epoxy to curing agent was 100:25. Finally, the epoxy composites containing the AgNW/GO hybrid fillers were cured at 120 °C for 8 h.

### 2.5. Material Characterization

The crystal structure of AgNWs and GO nanosheet were determined by X-ray diffractometer (XRD; Shimadzu XRD-6000, Kyoto, Japan) with Cu-Kα radiation. Morphologies of the nanofillers and their distribution throughout the matrix were observed using imaging techniques including scanning electron microscopy (SEM; Verios G4, FEI, Hillsboro, OR, USA) and transmission electron microscopy (TEM; Talos F200X, FEI, Hillsboro, OR, USA). Thermogravimetric analysis (TGA) of composites was conducted using a thermo-analyzer (Q5000IR, TA Instruments, New Castle, DE, USA) from 25 °C to 700 °C at a heating rate of 10 °C min^−1^ in air atmosphere. Melting behavior of the composites was investigated using differential scanning calorimetry (DSC; Q2000, TA Instruments, New Castle, DE, USA) in N_2_ atmosphere. Thermal conductivity of the composites was measured by laser flash analysis (LFA 447 NanoFlash, *NETZSCH*, Bavaria, Germany) and calculated from the following formula:
*k* = *α* × *ρ* × C_p_, 
where *k* is the thermal conductivity (W/mK), *α* is the thermal diffusivity (mm^2^/s), *ρ* is density (kg/m^3^), and C_p_ is the specific heat capacity (J/kg.K) of the composite. Volume electrical resistivities were measured by a volume resistance tester (LST-121, Beijing, China). Break strength was measured by a material testing machine (GNT100, Beijing, China). During the experiment, changes in temperature of composites were measured using an infrared camera (IRS-S6, Shanghai, China).

## 3. Results and Discussion

### 3.1. Morphology of AgNW and GO

The 1-D AgNWs were synthesized at 210 °C for 1 h using an alcohol-thermal synthesis method previously described by Yang and co-workers. Compared with ethylene glycol, glycerin contains more hydroxyl groups, so it has stronger reducibility than ethylene glycol and was used as the reducing agent. Successful preparation of AgNWs was confirmed by observing SEM micrographs of their morphology, as shown in Figure 2a. The purified AgNWs were found to be 1-D nanomaterials with length and diameter of approximately 15 μm and 120 nm, respectively. From the SEM image, we can see a random network structure composed of a large number of AgNWs, which can provide phonon transmission channels and effectively reduce thermal resistance. In addition, the microstructure of AgNWs was investigated by XRD. The five diffraction peaks appearing in the XRD pattern correspond to the (111), (200), (220), (311), and (222) planes of the silver crystal, respectively [41]. 

Morphologies of the GO nanosheets were directly observed using TEM, as shown in Figure 2b. Translucent GO nanosheets can be observed from the TEM image and have many wrinkles on the surface. During the synthesis of GO nanosheets, groups such as hydroxyl groups, carboxyl groups, and epoxy groups were introduced to the surface, which causes many wrinkles on the surface of the nanosheets. The crystal structure of graphite and GOs was characterized by XRD, and the results are shown in Figure 2c.

It can be seen from the spectrum that the diffraction peak of graphite appears at 26°, which is a typical (002) characteristic diffraction peak. We calculated that the spacing between adjacent graphite sheets was about 0.34 nm. In comparison, the (001) characteristic diffraction peak of GO nanosheets appeared at 10.7°, and the corresponding interlayer spacing was 0.83 nm. The layer spacing of the GO nanosheets after oxidation was increased due to the presence of a large number of functional groups on the surface [42].

### 3.2. Characterization of AgNW/GO Hybrid

The AgNW/GO hybrid was fabricated via a solution mixing method by simply mixing together the ethanol solutions of each filler, as shown in Scheme 1. The uniform states of AgNW and GO dispersions were broken down, indicating assembly of the AgNW/GO hybrid. The simultaneous occurrence of the precipitation processes demonstrated interaction between AgNWs and the GO nanosheets. Ultrasonication ensured that AgNW/GO hybrids were uniformly dispersed and fully assembled in the ethanol solution. The morphology of AgNW/GO hybrids was characterized by SEM and the results are shown in Figure 3. It can be seen that AgNWs fill the interlayer space of the GO nanosheets, resulting in a 3-D hybrid network. It is well known that fillers with different shapes affect the structure and performance of hybrid materials. Therefore, morphological changes in the resulting AgNW/GO hybrids with different percentages of AgNWs, from 15% to 90%, are presented in Figure 3. As the proportion of AgNWs increases, more AgNWs fill the regions between GO nanosheets, resulting in a 3-D layered network structure. The linear AgNW can connect adjacent GO nanosheets, effectively increasing the contact area between the layers. The 3-D network formed by AgNW and GO can provide better phonon transmission channels.

The 3-D network constructed by AgNW and GO nanosheets was further characterized by XRD and the results are shown in Figure 3. It can be seen that the diffraction peak of the GO nanosheets appears at 10.5°, indicating that GO sheets are not restacked after the addition of AgNWs. The AgNW network embedded in GO layers acts as a physical barrier to prevent restacking of the GO sheets during the formation of the 3-D hybrid network. Diffraction peaks of AgNWs also appeared in the hybrid fillers, verifying the presence of AgNWs.

Interactions between AgNWs and GO nanosheets were characterized by XPS. 

The C 1s XPS spectra of GO nanosheets and AgNW/GO hybrids, including the original curves and deconvolution results, are shown in Figure 4b,c. The deconvoluted peaks at 284.4, 284.9, 285.4, 286.5, 287.4, and 288.6 eV in the spectrum of GO correspond to C=C, C–C, C–OH, C–O–C, C=O, and O–C=O groups, respectively [43]. The C 1s XPS spectrum demonstrates the presence of a large number of functional groups on the surface of GO nanosheets. Compared with the deconvoluted peak of GO, the AgNW/GO hybrid shows a new peak at 285.9 eV, which corresponds to the C–N group. Moreover, the heights of the peaks of the C–C, C–OH, and C=O groups of the AgNW/GO hybrid are also increased. Changes in the C 1s spectrum of the hybrid can be attributed to residual PVP and glycerol on the surface of the AgNWs. 

The results of XPS analysis demonstrated that an interaction occurred between AgNW and GO during the formation of AgNW/GO hybrids. Figure 4d shows Ag 3d XPS spectra of AgNWs and AgNW / GO hybrids. The binding energy of Ag 3d in the spectrum of AgNW was 368.4 eV and 374.4 eV. However, the bonding energies of Ag 3d in AgNW/GO hybrids were 367.9 eV and 373.9 eV, which were shifted to a lower energy due to the transfer of electrons from metallic Ag to GO nanosheets, as a result of differences in the work functions of Ag (4.2 eV) and GO (4.48 eV) [44]. The Ag 3d XPS spectra also indicate the presence of interactions between AgNWs and GO nanosheets.

### 3.3. Microstructure of Composites

In general, acetone is used as the solvent during the curing of epoxy resin. However, due to the severe agglomeration of AgNWs in acetone, dispersion of AgNWs in the composites is poor. Therefore, ethanol was used to ensure uniform dispersion of fillers. In order to observe the dispersion of fillers in the epoxy, the fracture surface morphology of the composite was characterized by SEM, as shown in Figure 5.

Figure 5a shows the fracture surface morphology of pure epoxy, from which a smooth surface can be observed. However, compared to pure epoxy resin, the fracture surfaces of composites are relatively rough after the addition of AgNWs and GOs, as shown in Figure 5b and Figure 5c. It can be observed that AgNW and GO exhibit good dispersibility in epoxy and no agglomeration occurs. The good compatibility between fillers and the matrix is attributed to the functional groups present on the surface of AgNWs and GO nanosheets. The dispersion of the AgNW/GO hybrid in the epoxy is shown in Figure 5d, and it can be seen that the hybrid fillers with a 3-D network structure were uniformly embedded in the resin matrix. No large agglomeration of AgNWs or GOs was observed in the SEM image, and the hybrid fillers were well dispersed in the matrix. The uniform dispersion of fillers in the epoxy resin is beneficial for improving the performance of the composites.

### 3.4. Thermal Conductivity of Composites

Thermal conductivity of the composites was measured by laser flash analysis and calculated from the following formula:
*k* = *α* × *ρ* × C_p_, 
where *k* is the thermal conductivity (W/mK), *α* is the thermal diffusivity (mm^2^/s), *ρ* is density (kg/m^3^), and C_p_ is the specific heat capacity (J/kg.K) of the composite.

In order to investigate the relationship between the content of AgNW/GO hybrid fillers and the thermal conductivity of epoxy composites, the thermal conductivity of composites containing different proportions of fillers was measured. The results are shown in Figure 6. Furthermore, to investigate synergetic effects, the total concentration of each filler was maintained at a constant value of 4 wt.%. As a control, the thermal conductivity of composites containing AgNWs and GO nanosheets only were considered, 0.35 and 0.28 W/mK, respectively. The amount of AgNWs and GO nanosheets in the hybrid fillers were adjusted to vary the proportion of AgNWs from 15% to 90%. It can be observed from Figure 6a that the thermal conductivity of epoxy composites increased from 0.3 to 0.49 W/mK as the proportion of AgNWs increased from 15 to 75%. When the proportion of AgNWs was fixed at 75%, the thermal conductivity reached a maximum value of 0.49 W/mK. With further increases in the percentage of AgNWs, the thermal conductivity gradually decreased to 0.44 W/mK. 

However, the thermal conductivity of epoxy composites filled with AgNW/GO hybrid fillers was still higher than that of pure AgNW or GO.

By comparing the thermal conductivity of epoxy composites filled with AgNW/GO hybrids and the values of AgNW or GO alone, it can be confirmed that they have a synergistic effect in improving the thermal conductivity of epoxy composites. The synergy between AgNW and GO is attributed to the 3-D network structure formed by them, which can provide more efficient heat conduction channels. Gaps in the AgNWs network are filled with GO nanosheets, which increase the number of interfacial contacts between adjacent AgNWs, thereby forming more efficient heat conduction channels. In addition, GO nanosheets covering the surface of the AgNW network can effectively prevent the slow oxidation of AgNWs, thus ensuring that the thermal conductivity of the composite does not decrease in subsequent use. The relative proportions of AgNWs and GO nanosheets have a great influence on the thermal conductivity of epoxy composites. When the percentage of AgNWs was fixed at 75%, the thermal conductivity of epoxy composites reached a maximum value. When the GO content was low, gaps in the AgNW network were not effectively filled, resulting in higher thermal contact resistance between the fillers. 

The presence of excessive GO nanosheets in the network can lead to agglomeration and hinder the formation of effective heat conduction channels. Therefore, when the total content of filler is constant, the relative proportions of AgNWs and GO nanosheets in the hybrid filler are critical and selecting the right ratio can significantly improve the thermal conductivity of epoxy composites. We also investigated the relationship between the filler content and the thermal conductivity of epoxy composites when the relative proportion of AgNWs was maintained at 75%. It can be observed from Figure 6b that the thermal conductivity of epoxy composites exhibited an increasing tendency as the content of hybrid filler increased. Moreover, the thermal conductivity of epoxy composites filled with AgNW/GO hybrids was higher than the value of those with AgNWs or GO nanosheets alone. When the concentration of AgNW/GO hybrid filler was 10%, the thermal conductivity of epoxy composites was 1.2 W/mK, which was 64% and 36% higher than that of composites with AgNWs or GO nanosheets alone, respectively. In addition, it can be seen that the thermal conductivity of the epoxy composite exhibits a nonlinear relationship with the AgNW/GO content, suggesting that a percolation network has been formed at a low filler content. The large specific surface area of the AgNW/GO hybrid filler may contribute to the formation of this percolation network.

### 3.5. Electrical Properties of Composites

We measured the volume electrical resistivities of GO/EP, AgNW/GO/EP, and AgNW/EP composites. The results are shown in Figure 7. For epoxy composites filled with AgNW/GO hybrids, the relative proportion of AgNWs was maintained at 75%. The electrical resistivity of GO/EP composite decreased slowly after the addition of GOs. Compared with the resistivity of pure epoxy (4 × 10^15^ Ω cm), the value of GO/EP composite decreased by only three orders of magnitude after adding 10% GOs. This is mainly due to the good electrical insulation of GO, and its electrical resistivity is in the range of 10~10^5^ Ω cm [45,46]. Compared with GO, AgNW is an excellent electrical conductive material with an electrical resistivity of 7.5 × 10^−15^ Ω cm [27]. Therefore, the electrical resistivity of AgNW/EP composite sharply decreased after the addition of AgNWs. The electrical resistivity of composite containing 10% AgNWs has decreased by nine orders of magnitude compared to epoxy. However, after the addition of AgNW/GO hybrid fillers, the electrical conductivity of the composite decreased slowly, which is mainly due to the presence of GO nanosheets. Composites containing 10% AgNW/GO hybrid fillers still have a high electrical resistivity of 2.2 × 10^9^ Ω cm, meeting the insulation requirements of TIMs. The metallic AgNWs have extremely high electrical conductivity and can easily form conductive paths inside the composite, which is disadvantageous for the application of thermal interface materials. The presence of insulating GO in the AgNW network can effectively hinder the transmission of electrons between AgNWs, thereby reducing the electrical conductivity of the composite. 

### 3.6. Thermal Management Capabilities of Composites

To investigate the thermal diffusion of AgNW/GO/EP composites, all samples were placed on a heating platform at the same initial temperature. As samples were heated, the heating power was held constant to simulate the heat dissipation process of electronic devices during actual use. Surface temperatures of the nanocomposites were monitored by an infrared thermal imager and variation of surface temperatures with time as well as corresponding optical photographs are shown in Figure 8. From the curves, it can be seen that the surface temperature of AgNW/GO/EP composites increases faster with time compared to other materials. This is because the AgNW/GO/EP composite has a much better thermal response owing to higher thermal conductivity. Infrared thermal images taken at different times also visually demonstrate better heat dissipation performance of AgNW/GO/EP composites. Both the temperature curves and the infrared thermal images indicate that AgNW/GO/EP composites have excellent potential for thermal management applications.

### 3.7. TGA and DSC Analysis of Composites

Thermal stabilities of the composites were characterized using TGA. Figure 9a shows the weight loss curves of pure epoxy, AgNW/EP, GO/EP, and AgNW/GO/EP nanocomposites. Here, we use T_5%_ and T_50%_ to characterize the weight loss of the composites, representing the temperature at which the weight loss is 5% and 50%, respectively. Table 2 shows the TGA parameters of epoxy composites. The T_5%_ and T_50%_ of the epoxy resin were 271.8 and 372.1 °C, respectively. For epoxy composites the T_5%_ and T_50%_ slightly increased after the addition of AgNWs or GO nanosheets. However, the T_5%_ and T_50%_ of composites containing AgNW/GO hybrid filler increased significantly, by 344.7 °C and 415.9 °C, respectively. In other words, composites containing AgNW/GO hybrids demonstrate the highest thermal stability. 

Several positive factors related to the AgNW/GO hybrid filler are responsible for the significant increase in thermal stability of the AgNW/GO/EP composite. First, due to the presence of functional groups on the surface of AgNW/GO hybrids, the interfacial adhesion between the fillers and the epoxy resin is enhanced, thereby increasing the activation energy of thermal decomposition. In addition, fillers dispersed in the matrix act as a physical barrier, which greatly delays the transmission of thermal degradation products [47].

Figure 9 shows the DSC curves of pure epoxy and its composites. Compared with pure epoxy resin, the glass transition temperature (*T*_g_) of the AgNW/EP, GO/EP, and AgNW/GO/EP were increased by 7 °C, 10 °C, and 14 °C, respectively. The increased *T*_g_ can be explained by the following factors: First, nanofillers such as AgNWs and GOs have an extremely high specific surface area, resulting in physical crosslinking between the fillers and the epoxy molecular chains; secondly, a large number of functional groups present on the surface of the fillers have a strong interfacial interaction with the epoxy resin matrix. The data also suggests that composites containing AGNW/GO hybrid fillers have better thermal stability.

### 3.8. Mechanical Properties of Composites

In order to characterize the mechanical properties of the materials, we performed impact tests on pure epoxy, GO/EP, AgNW/EP, and AgNW/GO/EP composites. 

As shown in Figure 10, the impact strengths of these composites are 16.26 KJ/m^2^, 22.35 KJ/m^2^, 18.67 KJ/m^2^, and 28.85 KJ/m^2^, respectively. The impact strength of the AgNW/GO/EP composite was 77% higher than pure epoxy and also higher than values for the AgNW/EP or GO/EP composites. The increase in impact strength of AgNW/GO/EP composites clearly demonstrates the synergistic effects of the hybrid fillers. The synergistic effect of AgNW/GO/EP composites on mechanical properties can be explained by the following factors: First, the 3-D network formed by AgNWs and GO nanosheets can effectively prevent the agglomeration of GO nanosheets and increases the interface contact area between AgNW/GO hybrids and epoxy matrix.

Secondly, the AgNWs with high aspect ratio can be entangled with the epoxy molecular chains, thereby improving the interfacial adhesion between the AgNW/GO hybrids and the epoxy matrix. As a result of the synergistic effects between AgNWs and GO nanosheets within the epoxy matrix, the impact strength of composites is improved.

## 4. Conclusions

AgNW/GO hybrids with 3-D network structure were prepared using 1-D AgNWs and 2-D GO nanosheets. AgNW and GO show synergistic effects in improving the thermal conductivity and impact strength of epoxy composites. The thermal conductivity and impact strength of AgNW/GO/EP composites were improved compared to AgNW/EP or GO/EP composites. Moreover, the thermal decomposition temperature and glass transition temperature were also improved. During preparation of hybrid fillers, the 3-D network structure formed by the AgNWs and GO nanosheets can effectively increase the area of contact between each filler element. The large contact area decreases interfacial resistance within the hybrid fillers. The GO nanosheets were tightly embedded in the AgNW network structure, filling empty spaces. This resulted in the formation of a 3-D hybrid network with larger contact areas, the formation of more effective phonon transport channels, and enhanced the interaction between the fillers and epoxy matrix. Moreover, the presence of AgNWs can prevent the aggregation of GO nanosheets, thereby increasing the dispersion of filler within the epoxy matrix. We believe the hybrid filler system consisting of 1-D AgNWs and 2-D GO nanosheets will be an important method for improving the performance of composites in the future.

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
