# Peer review of "Synergetic Effects of Silver Nanowires and Graphene Oxide on Thermal Conductivity of Epoxy Composites"

_nanomaterials, 2019, doi:10.3390/nano9091264_

Round 1
Reviewer 1 Report
1) The authors wrote “Novel two-dimensional (2-D) graphene oxide (GO) has attracted a great interest due to its extremely high aspect ratio and high thermal conductivity.” In reality, GO has rather low thermal conductivity compared to other carbon fillers. Its attractiveness for thermal applications mostly comes low cost and mass production capabilities. I am guessing that mechanical properties of GO may also be useful for forming the percolation networks. The authors should correct their statement, add values of the thermal conductivity of GO, compare them with the base material thermal conductivity and provide references to papers that have values for GO, graphene and other carbon (relevant refs: “Phonons and thermal transport in graphene and graphene-based materials,” Reports on Progress in Phys., vol. 80, no. 3, p. 36502 (2017).
2) “Recently, one-dimensional (1-D) silver nanowires (AgNWs) have emerged as promising fillers for thermally conductive materials due to their high thermal conductivity and large specific surface area.” Provide the values of thermal conductivity of these nanowires and relevant reference. Indicate if the thermal conduction due to electrons or phonons.
3) What are the electrical properties of the composites? Many applications require heat conduction but electrical insulation. Is the percolated network mostly due to nanowires (and then it is electrically conductive) or it is not? The authors should address this issue. Do you see the super-linear dependence of the thermal conductivity on the filler content to claim that the percolation network is formed?
4) There have been several recent papers on papers with graphene-metallic, graphene-BN and other carbon derived fillers, which reported strong increase in the composite thermal conductivity (see F. Kargar, et al., "Thermal percolation threshold and thermal properties of composites with high loading of graphene and boron nitride fillers," ACS Appl. Mater. Interfaces, vol. 10, no. 43, pp. 37555–37565, 2018; F. Kargar, et al., Adv. Electron. Mater., vol. 5, no. 1, p. 1800558, 2019; J. S. Lewis, et al., “Thermal and electrical conductivity control in hybrid composites,” Mater. Res. Express, vol. 6, no. 8, p. 085325, 2019; Zahra Barani, et al., “Thermal Properties of the Binary-Filler Composites with Few-Layer Graphene and Copper Nanoparticles”, arXiv:1905.08725 (2019) – carbon – metal fillers). These prior papers should be cited and the results compared. This will help to place the present manuscript in the general context of thermal composite research and assess the usefulness of GO – metal nanowires fillers compared to other fillers.
Reviewer 2 Report
In the present manuscript, the authors reported the preparation of an epoxy composite based on the synergic effect between the filler composed by Ag nanowires (AgNW) and graphene oxide (GO). The 10 wt% AgNW/GO hybrid fillers were added in solution and provided thermal conductivity of 1.2 W/mK and the impact strength was 28.85KJ/m2, i.e. values higher than epoxy composites prepared with AgNWs or GO alone. The authors well reported the characterization of the fillers alone and their respective mixture at different content. Nevertheless, I suggest the authors to add more details on the epoxy matrix characterization. No detail of their chemical components were reported nor discussion about the possible influence of the filler on the crosslinking degree. Moreover I suggest the authors to compare the results gathered from thermal conductibity and mechanical properties with those of the main literature on similar materials. Overall, I consider the manuscript suitable for the journal and I suggest the authors to revise their paper before resubmission.
Round 2
Reviewer 2 Report
The authors have addressed most of the comments raised. The paper can be accepted for publication after adding some references in the introduction when authors describe the properties and applications of epoxy resins. Then a final check of the English is also suggested.
